

# Quantifying the spread in Sudden Stratospheric Warming wave forcing in CMIP6

Verónica Martínez-Andradas[1,2], Alvaro de la Cámara[1], Pablo Zurita-Gotor[1,2], François Lott[3], and Federico Serva[4]

[1]Dpto. Física de la Tierra y Astrofísica, Universidad Complutense de Madrid, Madrid, Spain
[2]Instituto de Geociencias (IGEO) UCM-CSIC, Madrid, Spain
[3]Laboratoire de Météorologie Dinamique, École Normale Superieure, Paris, France
[4]Consiglio Nazionale delle Ricerche, Institute of Marine Sciences (CNR-ISMAR), Rome, Italy

**Correspondence:** Verónica Martínez-Andradas (vemart05@ucm.es)

**Abstract.** Sudden stratospheric warmings (SSWs) show large spread across climate models in characteristics such as frequency of occurrence, seasonality and strength. This is reflective of inherent model biases. A well-known source of inter-model variability is the parameterized gravity wave forcing, as the parameterization schemes vary from model to model. This work compares the simulation of boreal SSWs in historical runs for seven high-top Climate Model Intercomparison Project Phase 6 models and in two reanalyses. The analysis is focused on the evolution of the different terms in the Transformed Eulerian Mean zonal-mean zonal momentum equation. A large spread is found through models and reanalyses in the mean magnitude of the resolved and parameterized wave forcing and the responses (wind deceleration and anomalous residual circulation). The results reveal that, in the stratosphere, both the wind deceleration and the strengthening of the residual circulation during SSWs correlate linearly across the models with anomalies in the resolved wave forcing. In the mesosphere, the forcing is a combination of resolved waves and, predominantly, parameterized gravity waves. Models with larger gravity-wave forcing anomalies produce larger changes in the residual circulation, while models with larger resolved wave forcing anomalies produce stronger wind deceleration, which we attribute to differences in the spatial shape of resolved and parametrized wave forcing. However, the forcing-response relation across events in the stratosphere is similar for each model, but not in the mesosphere. Our results are useful for interpreting the spread in projections of the dynamical forcing of SSWs in a changing climate.

## 1 Introduction

Temperatures in the stratosphere have been observed to decrease in recent decades in response to increasing greenhouse gases concentrations (Randel et al., 2016; Maycock et al., 2018), and are consistently expected to continue doing so in the future (Lee et al., 2021). The dynamical response of the wintertime polar vortex in the Northern Hemisphere is, however, much more uncertain (Manzini et al., 2014; Karpechko et al., 2022). Using different scenarios of increasing $CO_2$ in experiments of the Climate Model Intercomparison Project Phases 5 and 6 (CMIP 5 and CMIP6, respectively), Karpechko et al. (2022) concluded that half of the model spread in the future vortex mean state is related to model uncertainties. The projected changes



in vortex variability also suffer from large uncertainty, with individual models producing robust changes of different signs in the frequency of sudden stratospheric warmings (Ayarzagüena et al., 2018; Ayarzagüena et al., 2020).

This model spread in the representation of the vortex is not only present in future climate projections but also in simulations of the present climate, i.e. historical runs. In this context, understanding the inherent multi-model spread in the polar vortex climatology and variability is important.

It is known that the frequency of sudden stratospheric warmings (SSWs) in CMIP6 models differs from that observed in reanalyses (Ayarzagüena et al., 2020; Hall et al., 2021). This has been linked to biases in the climatological strength of the polar vortex and the upward wave activity fluxes in the lower stratosphere (Wu and Reichler, 2020). Regarding the SSW type, Hall et al. (2021) found an underestimation of split events in CMIP6, related to vortex geometry and to high filtering of wavenumber 2 and higher waves because of a stronger lower stratospheric vortex. However, Zhao et al. (2022) found that although the spatial pattern of the polar vortex is well reproduced in CMIP6 models, it is weaker than in reanalysis, which could be linked to less accumulated wave activity flux in the stratosphere in models. The main focus of these studies has been on the simulated frequency of SSWs in models, and much less attention has been given to quantifying the wave forcing of SSWs and the response of the stratospheric circulation.

The zonal-mean dynamics during the development of SSWs is the following. The strong zonal wind deceleration is driven by the anomalous Rossby wave activity flux convergence through wave breaking in the winter stratosphere (Limpasuvan et al., 2004). Using ERA5 reanalysis Cullens and Thurairajah (2021) concluded that although SSWs main drivers are planetary Rossby waves, gravity waves also contribute to their occurrence. Also for the austral polar vortex breakdown, both resolved and parameterized gravity waves in ERA5 were found to be important (Gupta et al., 2021). Wave breaking in the extratropical stratosphere not only decelerates the vortex but also induces a strengthened poleward residual circulation (e.g., de la Cámara et al., 2018). This motion causes downward shrink and adiabatic heating over the polar stratosphere. Weakened westerlies allow more eastward gravity waves to propagate up to the mesosphere. Consequently, the mesosphere experiences anomalous positive gravity wave forcing (more eastward gravity waves), decelerating the mesospheric residual circulation (Liu and Roble, 2002), cooling it over the pole and sinking the stratopause down to lower levels (Von Zahn et al., 1998; Limpasuvan et al., 2012). Once the zonal wind reversal has taken place, stationary planetary waves can not propagate upward and the wave forcing rapidly decreases, allowing westerly winds to recover at upper levels while they stay weak in the lower stratosphere due to the longer radiative damping time scales (Newman and Rosenfield, 1997).

The relatively broad spatial resolution of climate models limits the wavelengths that can be explicitly resolved, and a large part of the gravity wave spectrum needs to be parameterized, even in high-resolution models. The parameterization schemes varies in each model giving rise to a wide variety of gravity wave representation and interactions with the resolved flow, which introduces a source of discrepancy among models.

In this work we analyze the model spread in resolved and parameterized wave forcing during SSWs in seven CMIP6 models and two reanalyses. We study, for the SSWs in each model, how the circulation responses change for different forcings and how this is reproduced in models. How much of the wave activity flux convergence is translated into advection of the residual circulation and how much into wind deceleration across models? How different are the contributions of resolved and parameterized



waves? The answer to these questions in historical simulations, comparable with reanalyses, will be valuable for understanding future projections of a changing climate.

The article is organized as follows: datasets characteristics and methods are described in sections 2 and 3. An analysis on
the spread in the representation of sudden stratospheric warmings in the datasets is presented in section 4, first focused on the mean representation of SSWs and then on their variability. Finally, the conclusions are summarised in section 5.

## 2  Data

In the present work we use seven high-top models in CMIP6 with historical (all-forcing simulation of the recent past) experiments, from 1850 to 2014 (to 2009 in some models) and one or three runs, depending on the model. They are listed in Table
1 with information about their resolution, model top and parameterization schemes on orographic (OGW) and non-orographic gravity waves (NOGW). These models provide the output of the contributions to the momentum balance in the TEM formulation as described in Gerber and Manzini (2016). For HadGEM3-GC31-LL and UKESM1-0-LL tendencies due to meridional and vertical momentum advection terms are not available and have been calculated from other outputs, possibly leading to numerical errors. Both models are developed by the Met Office so they share most of their components, e.g. gravity waves pa-
rameterization scheme. Also MIROC6 and MRI-ESM2-0 share NOGW parameterization. We highlight that CESM2-WACCM includes NOGW sources by convection and frontal systems and IPSL-CM6A-LR also includes NOGW due to precipitation (Lott and Guez, 2013) and from frontal systems via a spontaneous adjustment mechanism (De la Cámara and Lott, 2015).

| Model | Variant_id | Top | Levels | Resolution | OGWs | NOGWs |
|---|---|---|---|---|---|---|
| CESM2-WACCM | r[1,2,3]i1p1f1 | 4.5e-6 hPa | L70 | 288x192 | Scinocca and McFarlane (2000) | Richter et al. (2010), Beres et al. (2005) |
| GFDL-ESM4 | r[1]i1p1f1 | 0.01 hPa | L49 | 360x180 | Garner (2005) | Alexander and Dunkerton (1999) |
| HadGEM3-GC31-LL | r[1,2,3]i1p1f3 | 85 km | L85 | 192x144 | Vosper (2015) | Warner and McIntyre (2001) |
| IPSL-CM6A-LR | r[1,2,4]i1p1f1 | 80 km | L79 | 144x143 | Lott (1999) | Lott and Guez (2013), De la Cámara and Lott (2015) |
| MIROC6 | r[1]i1p1f1 | 0.004 hPa | L81 | 256x128 | McFarlane (1987) | Hines (1997) |
| MRI-ESM2-0 | r[1,2,3]i1p1f1 | 0.01 hPa | L80 | 320x160 | Iwasaki et al. (1989) | Hines (1997) |
| UKESM1-0-LL | r[1,2,3]i1p1f2 | 85 km | L85 | 192x144 | Vosper (2015) | Warner and McIntyre (2001) |

**Table 1.** CMIP6 models with historical experiments used and their characteristics. In column *Variant_id* brackets show the different runs used. The historical experiment covers from 1850 to 2014, except for GFDL-ESM4 and IPSL-CM6A-LR that only extends to 2009.

Additionally, we use reanalysis data for comparison. We use the Transformed Eulerian Mean data set from the MERRA-2 and ERA5 reanalyses, as presented in Serva et al. (2024). The products are obtained by computation from 6-hourly data
following Gerber and Manzini (2016). For MERRA-2 native vertical 72 levels to 1 Pa and native spatial resolution $0.5° \times 0.625°$ (Gelaro et al., 2017) is used for the computation. For ERA5 native vertical 137 levels to 1 Pa and $0.5° \times 0.5°$ grid is used, while native spatial grid is of $0.25° \times 0.25°$ (Hersbach et al., 2020). In both data sets the contribution of gravity wave 'drag'



(GWD) is provided as a single term, not separated into OGW and NOGW as in the models. Therefore, in this work we will analyse it together as a GWD. In MERRA-2 GWD is only available up to 0.1 hPa.

## 3 Methods

We identify boreal SSWs in every climate model and the two reanalyses. The detection of SSWs is based on the criterion proposed by Charlton and Polvani (2007). The onset day of the event (lag 0) is set when the zonal mean zonal wind at 60 °N and 10 hPa crosses zero, provided no other event has been detected in the previous 20 days. The generation of SSWs by different models varies in total frequency and month of the year (Ayarzagüena et al., 2020; Wu and Reichler, 2020; Hall et al., 2021). This can be due to different vertical resolution and lid's height (Wu and Reichler, 2020) and should be taken into account when comparing the models. Figure 1 shows the distribution of SSW occurrence through the extended winter in each model of the present study, with the relative frequency of detected SSWs over the total number of years in brackets. Except for IPSL-CM6A-LR and GFDL-ESM4, all models generate less SSWs than observed with both reanalyses. Reanalyses also slightly differ with each other as the datasets do not fully match in time, but they agree over the common period (1980-2021, see Serva et al., 2024). We can also see that there is no agreement in the monthly distribution between models and reanalysis, as shown in Ayarzagüena et al. (2020). While in reanalysis January and February are the months with the highest frequency of SSWs, in models we find a more uniform distribution throughout the winter, with some biases towards late winter.

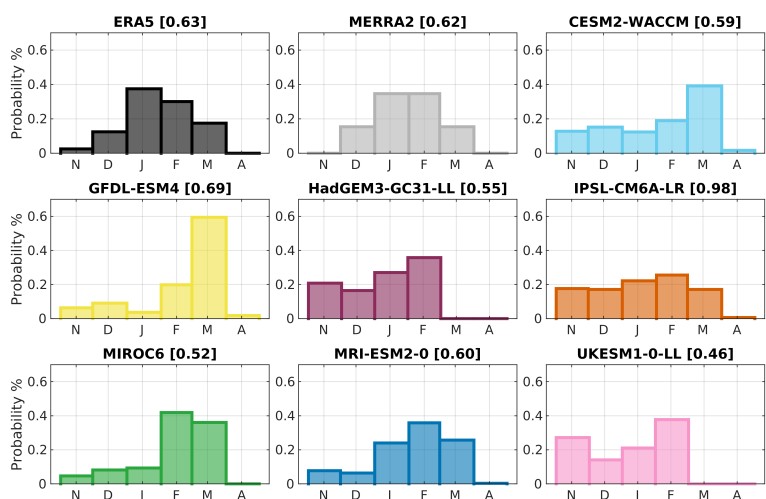

**Figure 1.** Frequency of SSW occurrence by month of the winter for each model and reanalyses. We show in brackets the SSW total frequency (number of events per year).





In this work we focus on analyzing the terms of the momentum balance equation in the Transformed-Eulerian Mean (TEM) formulation. Following Andrews et al. (1987), the meridional residual circulation in the TEM framework has a northward and a vertical component $(\overline{v}^*, \overline{w}^*)$, respectively. The TEM zonal momentum equation is then written as follows:

$$\overline{u}_t \; + \; \left\{ - \overline{v}^* \hat{f} + \overline{w}^* \overline{u}_z \right\} \; = \; \left\{ \; \frac{e^{\frac{z}{H}}}{a\cos\phi} \nabla \cdot \boldsymbol{F} \; + \; \overline{\chi}_{param} \; \right\} + \varepsilon \tag{1}$$

where

$$\hat{f} = f - \frac{1}{a\cos\phi}(\overline{u}\cos\phi)_\phi$$

and $f$ is the Coriolis parameter, $\boldsymbol{F}$ is the Eliassen-Palm flux, $\overline{\chi}_{param}$ is the parameterized physics forcing and $\varepsilon$ the residual, computed by the imbalance between both sides of the equation.

The terms in eq. 1 represent (from left to right) the following. On the left-hand side there is the temporal derivative of the zonal mean zonal wind (*Ut*) and in braces the advection by the residual circulation (*ADV*). This last term includes a northward and an upward contribution and the Coriolis effect. On the right-hand side the term in braces provides the total eddy-induced zonal mean force. This includes a contribution from resolved waves (*EPD*), basically from planetary waves, as the divergence of the Eliassen-Palm flux, and a contribution from parameterized waves (*GWD*), basically gravity waves.

The residual mean velocities $\overline{v}^*$ and $\overline{w}^*$ (defined in Andrews et al., 1987) may be expressed in terms of a stream function $\Psi$, where $\Psi$ can be calculated as:

$$\Psi(\phi, z) \; = \; \rho_0 cos(\phi) \int\limits_z^\infty \overline{v}^* e^{-\frac{z'}{H}} \; dz' \tag{2}$$

Finally, anomalies are calculated as the departure from the smoothed daily evolving annual cycle of each model, which is in turn defined based on a running 30-year mean. The statistical significance of the anomalies is assessed applying a two-tailed Student's t-test.

## 4 Results

### 4.1 SSW evolution in models and reanalyses

Before comparing the performance of different models, Figure 2 shows the composite evolution of the anomalies of the different terms in equation (1) during SSWs in ERA5, and displays well-known features of the life cycle of SSWs. There is a strong deceleration of the zonal-mean winds during the 7 days before the central date (Figure 2a), followed by an acceleration that is stronger in the upper levels than in the lower stratosphere, where the negative wind anomalies persist for at least one month. This contrasting evolution of the winds at different levels in the aftermath of SSWs is mainly the result of the faster radiative timescales at upper levels due to the different chemical composition (Hitchcock et al., 2013b).



The initial wind deceleration is driven by the negative resolved wave forcing anomalies with a peak at around 1 hPa and

lag -3 (Figure 2b). This deceleration ceases abruptly after lag zero and there even appear weak positive anomalies in the upper

stratosphere and lower mesosphere during the following 50 days. The parameterized forcing induces anomalous acceleration

mainly from lags -10 to 20 at mesospheric levels (above 1 hPa, Figure 2c), likely as a result of reduced eastward GW filtering

by the weak winds in the stratosphere below. Both the build-up and the disappearance of these anomalies is more gradual

than that of the resolved forcing. The positive anomalies at positive lags have been previously suggested to contribute to the

upper-level westerly wind recovery in the aftermath of SSWs (Hitchcock et al., 2013a).

The evolution of the residual mean momentum advection is determined by the combination of the resolved and parameterized

forcings (Figure 2d). At negative lags during the wind deceleration, there are negative anomalies in the stratosphere below 1

hPa (forced by the resolved forcing) and positive anomalies above (forced by the parameterized forcing); at positive lags the

evolution resembles that of the parameterized forcing.

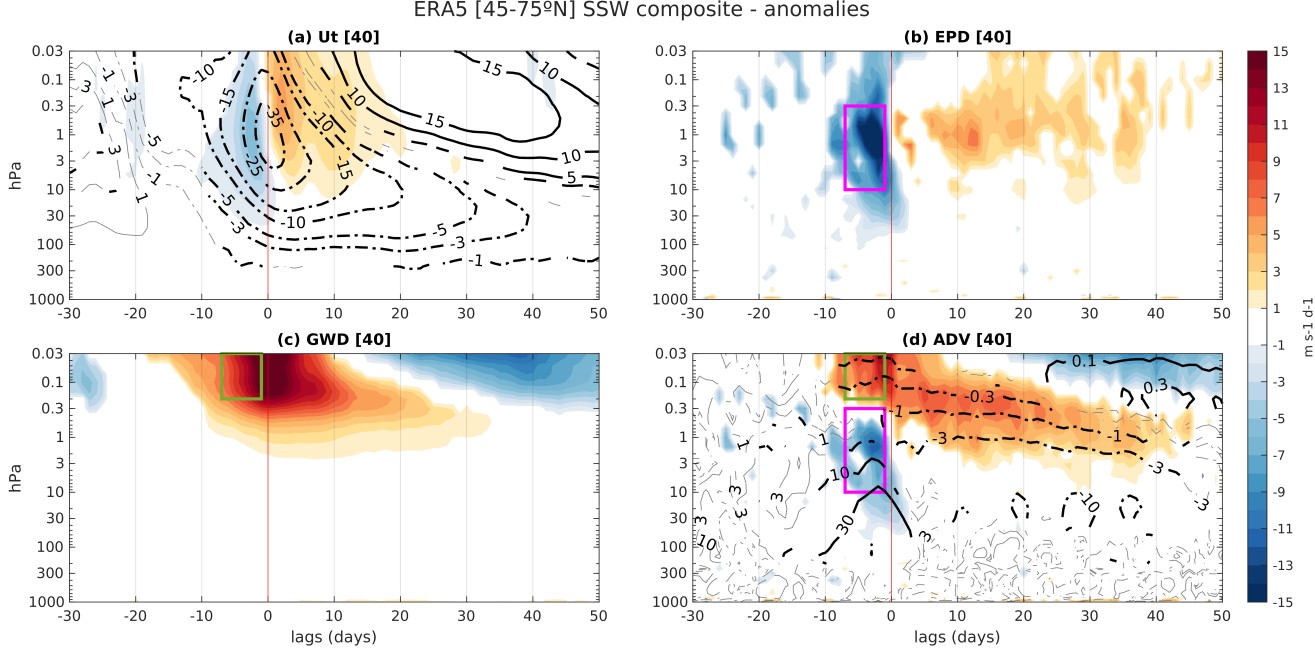

**Figure 2.** Lag-height SSW composite evolution of zonal mean [45-75°N] area averaged anomalies of (a) tendency of the zonal wind, (b) planetary Rossby waves 'drag', (c) gravity waves 'drag' and (d) the Coriolis torque of the residual circulation in shading for ERA5 reanalysis. Contours show anomalies of zonal wind in (a) and the mass stream function in (d). Units are $ms^{-1}d^{-1}$ for shading and $ms^{-1}$, $kgm^{-1}s^{-1}$ in (a,d) for contours, respectively. Shaded colors and black thick contours indicate values significant at 95 % confidence with a two-tailed t-test. The number of SSWs in the composite is shown in brackets. The pink (10-0.3 hPa, [45-75°N], -7 to -1 lags) and green (0.2-0.03 hPa, [45-75°N], -7 to -1 lags) boxes are used to average fields in subsequent analyses.





Similar figures to Figure 2 but for the climate models and MERRA-2 can be found in the supplemental material (Figures S1-S8), with qualitatively similar evolutions. To facilitate the inter-model comparison we show in Figure 3 the evolution of *EPD* (solid) and *GWD* (dashed) in the left column, and *Ut* (solid) and *ADV* (dashed) in the right column at some selected layers for the SSW composite mean anomalies. Each line represents the evolution of a different model mean.

We can see that there is agreement on the sign of the evolving anomalies among the models but with large inter-model spread, especially in the mesosphere. Different sponge layer thicknesses and biases in the climatologies (Figure 4) are likely to play a role for these model differences.

The spread in the mesospheric *GWD* anomalies around the onset date is comparable to the multimodel-mean, with values across models from 5 to 19 $ms^{-1}d^{-1}$ (see panels of [0.03-0.2 hPa]). ERA5 reanalysis and the models CESM2-WACCM and
MRI-ESM2-0 produce the largest *GWD* anomalies in the mesosphere, whereas GFDL-ESM4 and IPSL-CM6A-LR have the weakest.

The largest anomalies and spread in *EPD* are found in the upper-stratosphere-lower-mesosphere (see panels of [0.3-3 hPa]) around one week prior the onset date. Again a wide range of values can be found across models, varying from -5 to -20 $ms^{-1}d^{-1}$. In the mesosphere and stratosphere the anomalies are weaker, with less spread in the stratosphere.

The responses also show large spread across models. Prior to the onset date in the mesosphere the anomalies have opposite sign, as well as the forcings. Wind acceleration and *EPD* anomalies are negative at all levels, while *ADV* is positive in the mesosphere, as *GWD*, and negative below.

We argue that the in the stratosphere *EPD* drives both responses as *GWD* is relatively low. However, in the mesosphere positive *ADV* is driven by positive *GWD* and negative *Ut* by negative *EPD* anomalies. We come back to this point below.

**4.2    Model mean spread during the SSW development stage**

A burst of *EPD* and strong wind deceleration occurs approximately one week before the onset date (Figures 2 and 3). We will focus on this period for the subsequent analysis, referring to it as development stage. To characterize it we have selected -7 to -1 lags, as marked by the pink and green boxes in Figure 2. The sensitivity to the choice of these lags has been tested for some relevant results, as explained later. In this section we compare the SSW composite mean and quantify the biases across models
during the development stage.

During the development stage the main contribution to the total forcing (*EPD*+*GWD*) anomalies in the stratosphere is *EPD*, with much weaker *GWD* anomalies (Figure 4a). The spread through the models is notable, with the largest differences around 1 hPa. However, in the mesosphere the anomalous wave forcing is dominated by the reduction in the climatological westward *GWD* (Figure 4b), producing positive anomalies. The spread through the models is large in stratospheric *EPD* and mesospheric
*GWD* anomalies. Below 30 hPa the values converge. This also happens in the climatology (Figure 4b) with the spread growing with height.

Even though the total forcing anomalies reverse from negative in the stratosphere to positive in the mesosphere, the wind is decelerated at all levels, with maximum values occurring between 1 and 5 hPa depending on the model. In the mesosphere all models underestimate *Ut* compared to the reanalyses, as well as the climatological zonal-mean zonal wind. In contrast with *Ut*,





**Figure 3.** Evolution of *EPD* (solid) and *GWD* (dashed) in the left column and *Ut* (solid) and *ADV* (dashed) in the right column for the SSW composite mean. Each color represents a different model in CMIP6 and reanalysis. Rows show the average at different pressure layers. Variables are zonal mean and area-averaged [45-75°N].

*ADV* anomalies do reverse like the total forcing. While this change of sign occurs in all models, models differ on the height at which this happens.

One should note that the balance is not fully closed as there are still some residuals, even in the climatology. The MERRA-2 residual in particular stands out. This reanalysis has strong wind deceleration and anomalous residual advection in the mesosphere, but a comparable *EPD* to the rest of the models. This can only be compatible with very strong anomalous *GWD*,

but as this field is not provided above 0.1 hPa the residual is necessarily large. Similar outstanding residuals of eq. 1 in MERRA-2 are found in the tropical mesosphere by Ern et al. (2021).

To quantify the spread across models we separate between the stratosphere and the mesosphere based on the sign change. Taking into account that above 0.2 hPa all models produce positive *ADV* and forcing anomalies, we will characterize the





**Figure 4.** Vertical profiles of (a) SSW composite mean anomalies and (b) extended winter climatology of the models and reanalyses, [45-75°N] area averaged. Each color represents a different model in CMIP6 and reanalysis. Panels (from left to right) show the following variables: zonal mean zonal wind tendency/climatology, advection by the residual circulation & Coriolis effect, tendency of eastward wind due to Eliassen-Palm flux divergence, parameterized non-orographic and orographic gravity wave 'drag', the total wave forcing (sum of the previous two terms) and the residual errors in eq. 1. In (a) thick lines indicate 0.95 significance in the composite mean with a t-test. In (b) thick lines indicate significant differences from ERA5 reanalysis.

stratospheric and mesospheric anomalies in the following vertically averaging over the pink and green boxes in Figure 2, i.e.,

from 10 to 0.3 hPa and from 0.2 to 0.03 hPa respectively.

Using 'strat' and 'meso' to refer to the regions defined as stratosphere and mesosphere, Figure 5 shows various relations between the variables in Figure 4a vertically integrated as indicated above.





As expected from eq. 1, *EPD* in the stratosphere is linearly correlated with *Ut* and *ADV* (Figures 5 a,b). In both cases the correlation increases slightly when *GWD* is also included (not shown). The wind deceleration is also highly correlated with the advection in the stratosphere (Figure 5c).

In the mesosphere, the total forcing is a combination of *EPD* and *GWD*. We can see that the total forcing correlates well with *ADV* but not with *Ut* (Figure 5d,e). When looking at the forcings separately, we find that *GWD* is only correlated with *ADV* and *EPD* is only correlated with *Ut* (Figures 5f,i). The latter is consistent with the findings in Figure 3 (top) that the smaller *EPD* forcing plays a more important role than *GWD* for driving the zonal wind variability at mesospheric levels.

These results suggest that the *GWD* forcing is in quasi-steady balance with the residual advection, driving only a small part of the zonal wind variability. Because the total forcing is dominated by *GWD* but the zonal wind variability is driven primarily by *EPD*, there is no significant correlation between the total forcing and the wind tendency as noted above (Figure 5d).

The different character of the responses to *GWD* and *EPD* may reflect differences in the spatial structure of the forcings. The aspect ratio of a forcing determines how much of the response translates into zonal wind deceleration and how much into residual advection (Nakamura, 2024; Pfeffer, 1987). As shown in the Appendix in eqs. A1 and A2, *Ut* is forced by the second meridional derivative of the wave forcing, while *ADV* is forced by the second vertical derivative. Thus, shallow forcings will produce a stronger *ADV* response and deep forcings will produce a stronger *Ut* response. As shown in Figure 2, the planetary forcing *EPD* is deep, as it follows the polar vortex across the stratosphere and mesosphere, while the parameterized forcing *GWD* is shallow and concentrated at mesospheric levels (for more detail see latitude-height cross sections in the supplemental material, Figures S9-S17). This is consistent with their corresponding responses.

Another factor that may play a role for the different responses to *EPD* and *GWD* is their different time scales. As is also apparent in Figure 2 (and clearer in non-composited timeseries, not shown), *GWD* evolves more slowly than *EPD*, so we expect the response to *GWD* to vary in longer time scales than the response to *EPD*. We speculate that the different time scale of the two forcings may be due to the filtering effect of the stratospheric mean flow, essentially a time-integration of the stratospheric *EPD* anomalies.

Finally, Figure 6 shows that part of the spread across models can be explained by the inherent spread in their climatologies (Kim et al., 2017). Models with large climatological *EPD* values tend to also display large anomalies in the SSW composite. The same can be seen for the wind deceleration and the climatological zonal-mean zonal wind. Interestingly, IPSL-CM6A-LR produces large *EPD* climatological values while it has the weakest climatological zonal wind. The opposite occurs for MIROC6. In contrast, GFDL-ESM4 has both weak *EPD* and *Uzm* climatologies.

### 4.3 Inter-event variability in models during the SSW development stage

After having studied the biases between models in the mean SSW representation, we will now analyse the event-to-event variability in the forcings and circulation responses during the development stage of SSWs. We quantify how much of the resolved and parameterized wave deposition of momentum is translated into advection by the residual circulation and wind deceleration for every SSW simulated in the datasets.




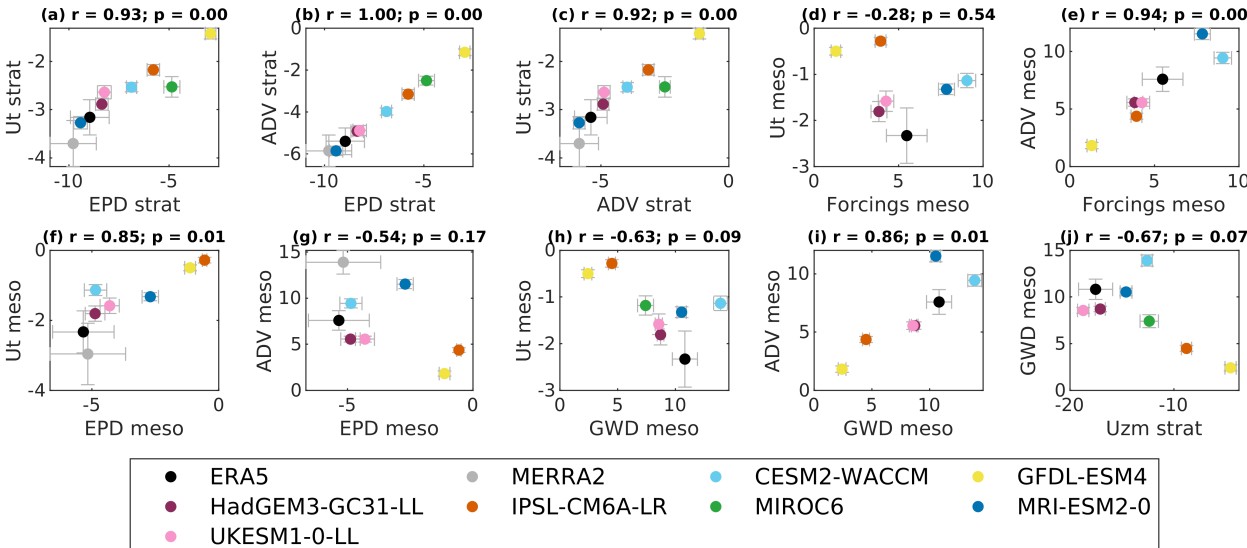

**Figure 5.** Scatter plots for the SSW composite mean of area-averaged fields at [45-75°N] and [-7 to -1 lags]. Variables are area-averaged from 10 to 0.3 hPa ('strat') and from 0.2 to 0.03 hPa ('meso'). Units are $\mathrm{ms^{-1}d^{-1}}$. Error bars show the standard error of the mean, taking into account all SSWs for each model and panel titles indicate the linear correlation coefficient (r) and p-value.

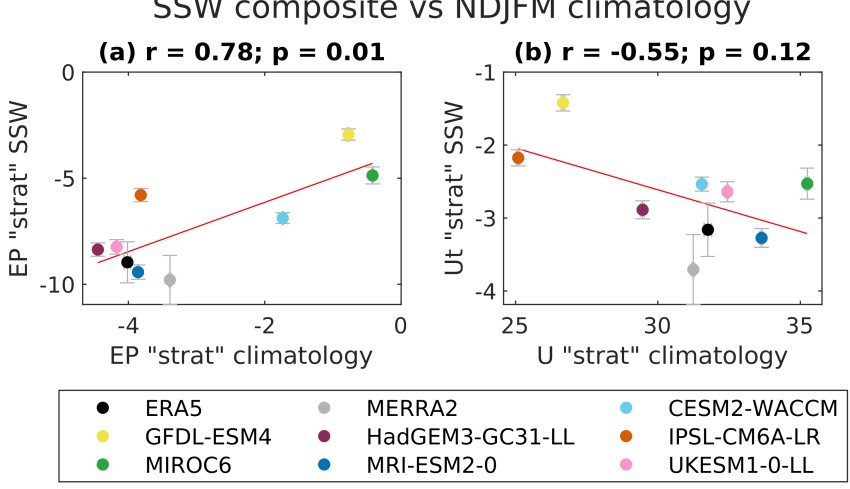

**Figure 6.** Scatter plots of SSW composite mean anomalies and climatology of *EPD* (a) and *Ut* (b) in the extratropical stratosphere (10 to 0.3 hPa and [45 to 75 °N]).

Starting with the stratosphere we quantify the relationship between the main forcing during the development stage, *EPD*, and the two responses, *ADV* and *Ut*. Since *GWD* is negligible in this region, it will not be taken into account. As expected



from eq. 1 the relation between *EPD* and the sum of *Ut* and *ADV* anomalies is completely linear (not shown). When separating the responses we also find linearity between *EPD* and both *ADV* and *Ut*, especially with *ADV*. This is shown in Figure 7a,

which also provides the linear correlation coefficient and the least-squares fit for both responses: $ADV \sim m1\,EPD + c1$; $Ut \sim m2\,EPD + c2$.

We find large correlation coefficients with values between 0.93/0.98 for *ADV* and 0.84/0.95 for *Ut*. The linear fit reveals that in both cases almost all models have similar slopes. This suggests that the dynamical relationship between both variables is similar for all models although there is some spread in the composite mean. Almost all models agree that approximately

2/3 of the *EPD* forcing is translated into *ADV* and 1/3 to wind deceleration, irrespective of the magnitude of the forcing. The exception is MIROC6, for which the response is translated equally into *ADV* and the deceleration of the wind. We can see that MIROC6 is also an outlier in Figure 5a . GFDL-ESM4 also differs for the *ADV* relation. This 2/3-1/3 ratio is roughly consistent with the findings of Nakamura et al. (2020) that about 60% of the wave forcing is balanced by residual advection and 40% is used to decelerate the zonal wind for ERA-Interim and MERRA-2 SSWs.

Moreover, we can see that for all models the slopes in both regressions are complementary so that they approximately add up to 1. A sensitivity test (Figure S18 in the supplementary material) shows that these slopes are robust when changing the lags used to average.

In contrast, we find that *EPD* is not linearly correlated with the wind anomaly (not shown). It is also interesting that no events with positive *EPD* anomalies are observed in the reanalyses, while they can be produced by the models.

In the mesosphere (Figure 8) the behavior is a bit more complex since gravity waves also play a role and the residuals are larger. However, the residuals do not seem to bias the relations as the correlation between the total forcings and both responses together is still close to one (not shown). The only exception is MRI-ESM2-0, which has the largest residuals (Figure 4a). Before discussing the relative impact of *EPD* and *GWD*, we first analyze how the response to the total forcing (*EPD+GWD*) is split between *ADV* and *Ut*. As in the previous section, we consider in Figure 8 averages from -7 to -1 lags, 0.2 to 0.03 hPa

and [45-75 °N]. Figure 8a shows that there is again a good correlation between the total forcing and *ADV*, though the scatter is larger than in the stratosphere. This may be due to higher inter-event variability in the partition of the response between *ADV* and *Ut*.

We also note that the slope of the fit can vary substantially across models. The fact that all slopes exceed 0.5 implies that *ADV* dominates the response to the forcing. However, this happens to different degrees in different models. In some models

(CESM2-WACCM, MRI-ESM2.0 and GFDL-ESM4) the slope is close to 1, suggesting that zonal wind variability plays little role in the response to the forcing for these models. Flatter slopes O(0.7) are found for ERA5 and IPSL-CM6A-LR, while the two MetOffice models have the smallest slopes, O(0.6). We have also analyzed the sensitivity of this linear fitting to the lags used for averaging (see Figure S19 in supplemental material). The slope for the *Ut* (*ADV*) fitting slightly decreases (increases) when extending the time-windows at negative lags. This is consistent with the transient nature of *Ut*: for long time averages

*Ut* should vanish and the full forcing be balanced by *ADV*. As shown in Figure 8a, in general the linear correlation between the forcing and *Ut* is better (and the slope of the fit is larger) for the models with smaller *ADV* slope, such as UKESM1-0LL





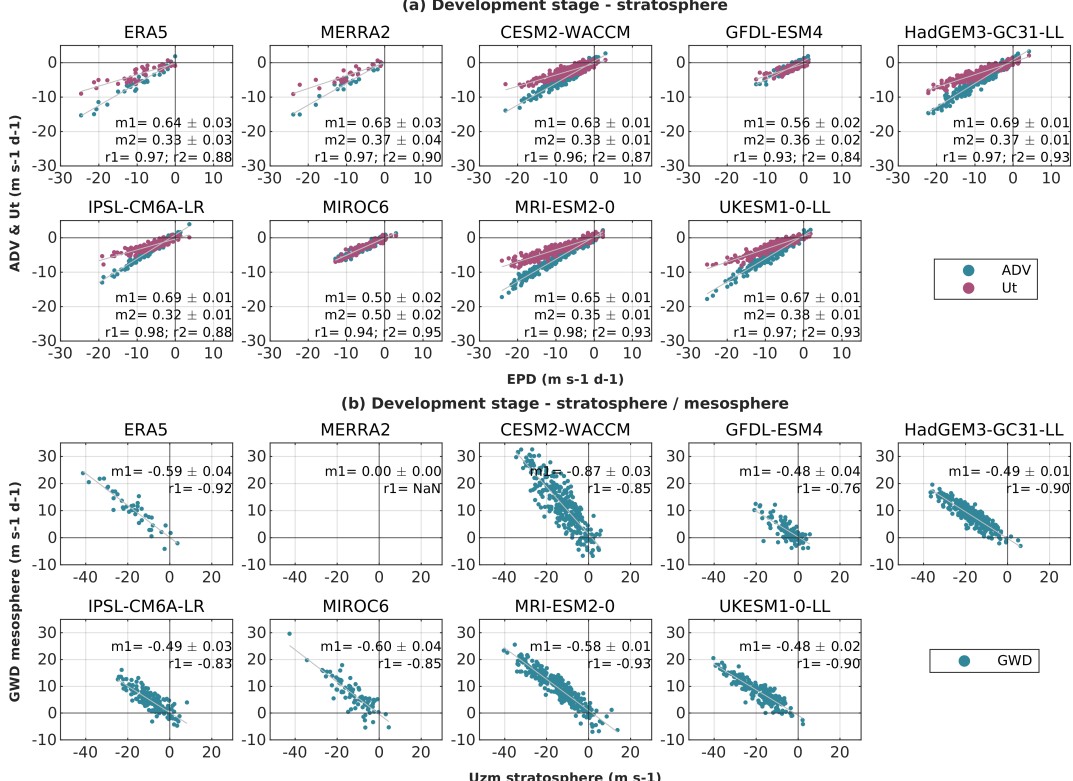

**Figure 7.** Scatter plots of the same variables in Figure 5 for all SSWs in the models and reanalyses. (a) *EPD* versus *ADV* and *Ut* in the stratosphere and (b) zonal mean zonal wind in the stratosphere versus *GWD* in the mesosphere. Stratospheric variables are averaged from 10 to 0.3 hPa and mesospheric variables from 0.2 to 0.03 hPa, and both are area-averaged from [45-75°N]. Linear fits $y \sim mx + c$ are also shown (grey lines), and the corresponding slopes (m1/m2 for *ADV* / *Ut* in (a) and m1 for *GWD* in (b)) are indicated in the text. The corresponding r-squared coefficients, r1 and r2, are also provided.

and HadGEM3-GC31-LL. In contrast, there is only noise in the scatter plot between the forcing and *Ut* for models such as CESM2-WACCM and GFDL-ESM4, for which *ADV* alone balances the response to the forcing.

 We next investigate the relative role of *GWD* and *EPD* for producing these responses (Figure 8b and c). Consistent with the

250 composite results, *EPD* is strongly biased to negative values and *GWD* to positive values, though there is significant inter-event variability (including values of the opposite sign) in most models. We recall that in the composite mean, *GWD* is the dominant forcing. Moreover, there is not linearity in the *GWD* versus *EPD* relation (not shown) pointing out that there is not a fixed balance between them. Most models produce a linear relation between *GWD* and *ADV* (Figure 8c), although the slopes of the fit varies substantially across models. In contrast, there is no linearity with *Ut*.

255 As regards *EPD*, we showed before that in the composite-mean this term is smaller than *GWD* in most models (though this small term was responsible for the zonal wind anomalies). When considering the intra-model variability, this is not necessarily



the case and *EPD* may be larger than *GWD* in some events and models. For those models with significant *EPD* variability, there is a weak correlation between *EPD* and both *ADV* and *Ut*, in general larger for the latter. It should be added that *EPD* and *GWD* are not correlated with each other (not shown).

Finally, we note the differences in the intercept for all these fits (values not shown). The regression lines (grey lines) nearly cross the origin for the *EPD/Ut* and *GWD/ADV* fits, but not for *EPD/ADV* and *GWD/Ut*. This means that the absence of *EPD* (*GWD*) anomalies does not imply the absence of *ADV* (*Ut*) anomalies.

These results together with the previous ones lead us to conclude the following. As the linear correlation with the responses is higher for the combination of *EPD* and *GWD*, both forcings contribute to the *ADV* and *Ut* anomalies but with different

proportions. *GWD* contributes more to *ADV* and *EPD* to *Ut*, possibly because of differences in their aspect ratios or time scales as discussed in section 4.2. However, this partition is highly model dependent as the slopes vary substantially across models. Additionally, the relatively high scatter indicates than the proportion also varies across the different SSWs in each model.

Finally, we investigate the relationship between the stratospheric wind and the mesospheric *GWD*. As shown in the previous section, the mean *GWD* anomalies are correlated with the wind profile beneath. When looking at the even-to-event variability

in these two variables (Figure 7b), we find a linear relationship with correlation coefficients over 0.76 in all models. Interestingly, the linear regression slopes vary across the models. This is related to the various schemes used in the parametrizations. In particular, CESM2-WACCM stands out for having a considerably different slope from all other models. This model is an outlier in Figure 5j. One possible explanation could be that The parametrization scheme used in CESM2-WACCM has interactive non-orographic gravity waves, i.e. some parameters in the gravity wave emission depend on other variables outside the

parametrization (Richter et al., 2010). However, IPSL-CM6A-LR also has an interactive scheme (De la Cámara and Lott, 2015) but the relationship is not different from other models, so the source may not be the reason. We also observe agreement across models that share the same parametrization, such as HadGEM3-GC31-LL and UKESM1-0-LL, or MIROC6 and MRI-ESM2-0 for the non-orographic waves.

## 5    Summary and Conclusions

This study performs an intercomparison of seven high-top CMIP6 models and ERA5 and MERRA-2 reanalyses in the representation of Northern Hemisphere sudden stratospheric warmings (SSWs). We focus on the terms of the zonal-mean zonal momentum balance in the Transformed Eulerian Mean formulation. Our analysis separates the wave forcing into resolved (*EPD*) and parameterized (*GWD*) waves, and investigates their effects on both the advection by the residual circulation (*ADV*) and the deceleration of the zonal mean wind (*Ut*).

First, we compare biases in the SSW composite mean across the models and reanalyses, finding a large spread in the anomalous forcing and circulation responses. Part of this spread is related to climatological biases in the vortex strength and planetary wave forcing. In the stratosphere, we find that models with larger *EPD* anomalies have larger *Ut* and *ADV* anomalies, with a high linear correlation across models. The contribution of parameterized gravity waves is negligible at this altitude. However, in the mesosphere the total forcing is a combination of resolved and parameterized waves. Models with larger *GWD*



anomalies produce larger changes in the residual circulation, while models with larger *EPD* anomalies produce stronger wind deceleration.

Second, we compare the inter-event variability in all the models, also focusing on the development stage of SSWs. We quantify how much of the wave forcing, resolved and parameterized, is translated into each of the responses. One important finding is that in the stratosphere the relationship between both responses and the resolved wave forcing is linear and quantitatively

similar in most models: approximately 2/3 of the anomalous *EPD* translates into anomalous *ADV*, and 1/3 into *Ut*, showing that the response of the residual circulation is larger than the wind deceleration. The large correlation coefficient indicates low inter-event variability in this result.

In the mesosphere, the response-to-forcing relation is also linear but with a larger spread across models, i.e. larger inter-event variability. At those heights the presence of sponge layers near the top boundary may be acting in many models together

with the dynamics. The total forcing primarily drives *ADV* anomalies, while the *Ut* response is noisy. But this relation is highly model dependent. We provide two possible explanations. First, we argue that the smoother evolution of *GWD* in the mesosphere should provide a stronger *ADV* response, while the pulse-like evolution of *EPD* primarily drives the transient wind deceleration. Another argument relies on the different aspect ratio of the two forcings (Nakamura, 2024), with the *GWD* anomalies being shallower.

We conclude that while there is a large model spread in the wave forcing of SSWs in historical CMIP6 integrations, the stratospheric response in most models is split in a similar proportion into zonal wind deceleration and meridional residual advection. In the mesosphere, the parameterized gravity wave forcing is dominant but the planetary forcing is also important, and the two forcings produce very different responses. Thus, for the same total forcing the response will be sensitive to how this forcing is partitioned into the resolved and parameterized components, consistent with the large variability seen in Figure

3. These results suggest that tuning the gravity wave parametrization in models to compensate for biases in the resolved waves and produce the same total forcing/climatological wind may produce the wrong temporal evolution during SSWs. It is hoped that the results presented here for historical simulations will also be useful for understanding the evolution of the SSW forcings in future climate projections.





**Figure 8.** As in Figure 7 but relating (a) the total wave forcing (*EPD+GWD*), (b) *EPD* and (c) *GWD* both with *ADV* and *Ut* in the mesosphere (0.2 to 0.03 hPa and 45 to 75 °N).



*Data availability.* Data of the momentum balance products in TEM formulation for the reanalyses are available in https://zenodo.org/record/
6959944 (Serva, 2022b) for MERRA-2 and https://zenodo.org/record/7081436 (Serva, 2022a) for ERA5.

## Appendix A

From the transformed-Eulerian mean zonal momentum equation (eq. 1) one can see that although the wave forcing decelerates the wind, part of this forcing is partially balanced by the Coriolis torque of the residual circulation. This equation alone does not provide enough information to assess the extent to which a given eddy forcing will decelerate the zonal wind. Following Nakamura (2024), one can use the thermal wind equation to derive the following equations for the slow, balanced evolution of the flow:

$$
\left( \frac{\partial^2}{\partial y^2} + \frac{1}{\rho_0} \frac{\partial}{\partial z} \left( \rho_0 \epsilon_0 \frac{\partial}{\partial z} \right) \right) \frac{\partial u}{\partial t} = \frac{\partial^2}{\partial y^2} \left( \frac{1}{\rho_0} \nabla \cdot \mathbf{F} + \overline{X} \right) - \frac{f_0}{\rho_0} \frac{\partial}{\partial z} \left( \rho_0 \frac{d\theta_0}{dz} \frac{\partial \dot{\theta}}{\partial y} \right),
\tag{A1}
$$

$$
\left( \frac{\partial^2}{\partial y^2} + \frac{1}{\rho_0} \frac{\partial}{\partial z} \left( \rho_0 \epsilon_0 \frac{\partial}{\partial z} \right) \right) f_0 v^* = -\frac{1}{\rho_0} \frac{\partial}{\partial z} \left[ \rho_0 \epsilon_0 \frac{\partial}{\partial z} \left( \frac{1}{\rho_0} \nabla \cdot \mathbf{F} + \overline{X} \right) \right] - \frac{f_0}{\rho_0} \frac{\partial}{\partial z} \left( \rho_0 \frac{d\theta_0}{dz} \frac{\partial \dot{\theta}}{\partial y} \right),
\tag{A2}
$$

where $\epsilon_0(z) = \frac{f_0^2}{N_0^2(z)}$, $\theta_0$ is the background potential temperature and $\dot{\theta}$ is the zonal-mean non-adiabatic heating. Eqs. A1 and A2 show that the second-order derivatives of the forcing drive the zonal acceleration (horizontal derivative) and the residual circulation (vertical derivative). Consequently, the aspect ratio of the forcing will determine the partition of the response into the acceleration of the zonal-mean zonal wind and the Coriolis torque of the residual circulation (Pfeffer, 1987; Nakamura and Solomon, 2010).

*Author contributions.* Alvaro de la Cámara and Pablo Zurita-Gotor conceived of the presented idea and designed the project. Verónica Martínez-Andradas performed the computations with the guidance of Alvaro de la Cámara, Pablo Zurita-Gotor and François Lott. Federico Serva provided the reanalyses data. Verónica Martínez-Andradas took the lead in writing the manuscript with the help of Alvaro de la Cámara and Pablo Zurita-Gotor. All authors provided critical feedback and helped shape the research, analysis and manuscript

*Competing interests.* The authors declare that they have no conflict of interest.

*Acknowledgements.* This work has been funded by the Spanish Research Agency project DYNWARM (PID2022-136316NB-I00). Author Martínez-Andradas has been supported by a predoctoral research fellowship funded by the Spanish Ministry for Science and Innovation (PRE2020-091812).



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
