# Peer review of "Quantifying the spread in Sudden Stratospheric Warming wave forcing in CMIP6"

_EGUsphere, 2024_

## Author Comment (AC1)

**RC1**

Minor comments:

1. In the abstract on the line 12, the sentence 'However, the forcing-response relation across events in the stratosphere is similar for each model, but not in the mesosphere.' can be confusing. Possibly sentence such as 'Although the forcing-response relation across individual SSW events is similar for each model in the stratosphere, this does not hold in the mesosphere.' would be clearer?

   Thank you for the suggestion. We have changed the sentence to the proposed one.

2. In figure 1 you show calculated SSW frequencies based on criterion by Charleton and Polvani (2007). Do you think other approaches (e.g. vortex moments as in Seviour et al., 2013) would give significantly different values and dates, which could in turn change the analysis of the composites? Since the used criterion uses strictly set thresholds – reversal of the zonal mean zonal winds at 60°N at 10 hPa – it is quite limiting (which is generally true for most of the SSW definitions). It could be interesting to provide second figure with set of frequencies based on another criterion just to illustrate how much the results do/do not depend on the chosen method.

   There are many proposed criteria to define SSWs and some studies have compared them (Butler et al 2016, Palmeiro et al 2016), showing that each definition has its pros and cons. This particular criterion is very simple and easy to implement to compare many simulations, and it is widely used, so it puts our results in context with other studies.

   We agree about the limitations of this criterion. Since it uses a set threshold, the anomalies and occurrence of SSWs are affected by the inherent biases in the models' climatologies, i.e. a model with a very strong vortex necessarily requires stronger decelerations and, consequently, stronger wave forcing to reach the zero line. However, one of the advantages of this criterion is its connection to the zonal mean wind, a variable of dynamical significance. After the onset date, the zero wind line prevents the vertical propagation of Rossby waves and eliminates the filtering of eastward gravity waves ($c>0$) –the same dynamics operate in all models even if the magnitude of the anomalies may be different.

However, we agree that other criteria could be tested to assess the robustness of the results presented in the manuscript. The method proposed by the reviewer (vortex moments, as in Seviour et al., 2013) has the inconvenience of requiring 2D data, which is complicated when using a large number of models. Instead, we propose to use the sudden stratospheric deceleration (SSD) criterion as in Birner and Albers (2017). This criterion is based on the 10-day wind change ($U_t$) dropping below two standard deviations at 60ºN, 10hPa. As this threshold is model independent it varies from -30 to -18 m s-1 10-day change.

| ERA5 | MERRA2 | CESM2-WACCM | GFDL-ESM4 | HadGEM3-GC31-LL | IPSL-CM6A-LR | MIROC6 | MRI-ESM2-0 | UKESM1-0-LL |
|------|--------|-------------|-----------|------------------|---------------|--------|------------|--------------|
| -30.1 | -28.6 | -20.8 | -18.3 | -23.4 | -24.5 | -23.0 | -25.5 | -23.3 |

Next, we discuss how some results of the article change for SSD. The frequency of occurrence of SSD increases considerably, by about a factor of two, with respect to SSW, except for IPSL-CM6A-LR (Figure R1). In models like HadGEM3-GC31-LL and UKESM1-0-LL there is low frequency both of SSWs and SSDs. IPSL-CM6A-LR has the most frequency of SSWs but relatively low SSDs, possibly indicating that the variability is low and the climatological vortex is weak. On the contrary, MIROC6 has few SSWs but many SSDs, consistent with a strong climatological vortex, i.e. very difficult to reach the 0 line, and with a lot of extreme variability. Finally, CESM2-WACCM, GFDL-ESM4, MRI-ESM2-0 have similar SSWs and SSDs compared to reanalysis.

The monthly distribution also changes. Generally, there are fewer SSDs in November and March.

[Figure]

Figure R1. As Figure 1 in the manuscript but using the sudden deceleration criterion (Birner and Albers 2017).

In the following, we discuss EPD, GWD, Ut and ADV anomalies in SSD composites.

[Figure]

Figure R2. As Figure 3 in the manuscript but for SSDs composites.

Looking at the evolution in Figure R2 we decided to compare the development stage of the article with [-3, 3] lags. So the SSD composite mean in the stratospheric and mesospheric selected regions shows the following scatterplots of the fields (Figure R3). There is linearity in the stratosphere between EPD and both ADV and Ut; and in the mesosphere between EPD/Ut and GWD/ADV.

[Figure]

Figure R3. As Figure 5 in the manuscript but for SSDs composites

These results are similar to Figure 5 in the manuscript but with a somewhat larger scatter. We conclude then that our results are robust independently of the criterion used.

3. On the line 110 you describe how the anomalies are calculated. Could you please specify how the annual cycles were smoothed? Also, did the 30-year mean include the dates where the SSW was happening? If yes, this way the mean values could be influenced enough to result in lesser anomalies in cases of models with high SSW frequencies (especially IPSL-CM6A-LR).

To compute the anomalies we subtract the annual cycle. This annual cycle is calculated using the surrounding 30 years ( 30-year running mean), so it evolves with time. This has the advantage of removing long-frequency trends. Additionally, the daily annual cycle is smoothed with a 30-day running mean.

As you point out, years with SSWs are also included in the climatology. Taking only years without SSWs may not be an option for models like IPSL-CM6A-LR with a very high frequency of SSWs. It is true that when comparing anomalies, these are subject to model biases, including the frequency of SSWs. This is explicitly discussed in Figure 6, which shows that the strength of the anomalies is indeed dependent on the model's climatology, making it ambiguous to differentiate between biases in the climatology and in SSW anomalies. However, because the linear scalings in Figure 5 are robust across models with different climatologies and anomalies, we expect our conclusions not to be affected by the specifics of the climatology definition (e.g., if we had used a non-SSW climatology for IPSL, we would expect the anomalies for this model to be bigger but still aligned)

Technical comments

1. In Figure 4, parts of lines which denote significance or difference from ERA5 are too thick to properly distinguish in some places. Especially in plot b), where ERA 5 is then almost invisible. Possibly you could make the lines thinner and at places, where the values are not significant (a) or significantly different from ERA5 (b), you could make the lines dashed?

Thank you for the comment. The lines have now been changed so that non-significant values are represented by dots. We are aware that in some parts of the

figure it is not easy to distinguish all the lines. However, where the values converge it is not as important to distinguish them from each other as it is to notice the dispersion where they do not. We hope that the figure is now clearer.

2. In Figures 7 and 8, some of the coefficients are hard to read. Maybe a table with those coefficients could be provided to summarise the results of these plots?
We agree that those Figures are hard to read because of the amount of information. We have improved them by clearing the text background to make it more readable.

3. I noticed that in several figures there are values missing for model MIROC6. After looking at figure S6 in the supplement I can see that there are missing values for ADV and EPD near the top of the model. I did not find any reference to this in the text. Could you please provide some information regarding this?
The data provided by the ESGF repository have NaN values above 0.07 hPa in the products of the wind tendency due to v*, w* and EP flux divergence (see the green line in Figure 4). After giving this some consideration, we have decided to include this model in this analysis averaging only up to 0.07 hPa. Thank you for noticing that this was not mentioned in the text, now we explain this issue.

4. On lines 252 and 258 there are two sentences concerning relationship between GWD and EPD which provide similar information I believe.

- 'Moreover, there is not linearity in the GWD versus EPD relation (not shown) pointing out that there is not a fixed balance between them.'

- 'It should be added that EPD and GWD are not correlated with each other (not shown).'
Thank you for noticing, I have removed the second sentence.

Comment/question regarding possible extension of the study:

As you mention in your manuscript, it has been observed, that split SSWs tend to be underestimated in models (Hall et al., 2021), which is connected to geometry of the polar vortex and its strength in the lower stratosphere. Have you considered dividing the data based on the type of SSW before the analysis? It could be interesting to see if the found relationships differ based on the type.
This is an interesting question: we agree it could be useful to distinguish between wave 1 and 2. However, 2D fields (not currently available to us) would be necessary for this analysis so this is beyond the scope of the present work. But we will consider this in future work.

References:

Charlton, A. J., and L. M. Polvani, 2007: A new look at stratospheric sudden warmings. Part I: Climatology and modeling benchmarks. J. Climate, 20, 449–469, doi:10.1175/JCLI3996.1.

Seviour, W. J. M., D. M. Mitchell, and L. J. Gray, 2013: A practical method to identify displaced and split stratospheric polar vortex events. Geophys. Res. Lett., 40, 5268–5273, doi:10.1002/grl.50927

Hall, R. J., Mitchell, D. M., Seviour, W. J., and Wright, C. J.: Persistent model biases in the CMIP6 representation of stratospheric polar vortex variability, Journal of Geophysical Research: Atmospheres, 126, e2021JD034 759, 2021.

Birner, T., & Albers, J. R. (2017). Sudden stratospheric warmings and anomalous upward wave activity flux. *Sola*, *13*(Special_Edition), 8-12, doi: https://doi.org/10.2151/sola.13A-002.

**RC2**

Comments:

L42: downward shrink? Maybe downward flow, or subsidence?

Thank you for noticing, it was a language misunderstanding. Changed now to downwelling.

L73: What is the period used for the reanalyses?

The periods are 1959-2021 for ERA5 and 1980-2021 for MERRA-2. Now this is included in the text.

L88 and Figure 1: can you really statistically distinguish, for example, 0.60 in MRI and 0.62 in MERRA? Can you provide some statistical estimates on the effect of sampling?

To have some statistical estimates we have performed a test proposed by Gu et al. (2008) that compares the ratio of the rate of the two Poisson processes. With this parametric test IPSL-CM6A-LR is the only model with significantly different frequency at a 95% confidence level. We have included this in the text.

Also, the frequency on y-axis is relative with respect to the total number of SSW simulated by particular model, right? Their add up to 1? I would specify this clearly. The alternative is the total fraction per each month, so that they would add up to the total frequency per season.

That is right, the frequency per month is relative to the total number of SSW in each model. This is now clarified in the text.

L103: Sorry for naïve question but why do you attribute the vertical advection to the Coriolis effect? I would not think of w_star*du/dz as of a Coriolis effect.

There is a misunderstanding in this sentence. The term f_hat in equation (1) includes the Coriolis effect and the latitudinal derivative. So v_star multiplied by f_hat represents the northward advection and the Coriolis effect. And then the term of w_star is the upward advection.

L119: "different chemical composition" could you clarify this statement? I would naively assume that the altitude must play a role too because the radiative damping should be related to optical depth, which in turn depends on altitude even if the chemical composition were homogenous throughout the atmosphere.

Thanks for the comment. The different chemical composition is important for the absorption and radiative damping but, as you pointed out, it is also dependent on the density and optical depth. We finally decided to remove the statement.

L211-224: I think you need to be explicit about your math. How do you get the 2/3 and 1/3 partitioning? By comparing m1 and m2? Are the values normalized? Please explain it. L225-227: Related to the previous question – why should m1 and m2 add up to 1?

The momentum balance Eq. 1 implies that ADV+Ut=EPD, neglecting the residual. A plot of ADV+Ut against EPD shows this to be very well satisfied: there is little scatter about a linear fit with a slope of 1 (not shown). Fig. 7 shows that the two responses ADV and Ut also scale roughly linearly with the forcing EPD, i.e., ADV = m1 EPD + c1; and Ut = m2 EPD + c2. Introducing these fits in the full balance leads to m1+m2=1. Their ratio is roughly ⅔ to ⅓ (m1= 0.63 = ⅔, m2= 0.36 = 1/3).

Note that while we would expect ADV and Ut to correlate positively with EPD, the fact that they do it linearly is not obvious but a major finding of this paper. Our results show that EPD is balanced in a roughly fixed proportion (also robust across models) by the ADV and Ut responses. This was not obvious a priori; for instance, one could have envisioned cases that one or the other response might dominate depending on the event. Our results show that this does not happen.

L252: There is NO linearity…
        Thank you. Changed.
L273: The –> the (no need to capitalize)
        Thank you. Changed.
L276: agreement in what?
        There is agreement in the slopes of the fittings. Now this is specified in the text.

References:

Gu, K., Ng, H. K. T., Tang, M. L., & Schucany, W. R. (2008). Testing the ratio of two poisson rates. *Biometrical Journal: Journal of Mathematical Methods in Biosciences*, *50*(2), 283-298.